# A Feasibility Study of Profiting from System Imbalance Using Residential Electric Vehicle Charging Infrastructure

**Marián Tomašov [1,2], Milan Straka [1,3], Dávid Martinko [4], Peter Braciník [1,2] and Ľuboš Buzna [1,3,5,*]**

[1] INO-HUB Energy j.s.a., Tomášikova 30, 821 01 Bratislava, Slovakia; marian.tomasov@inohub.sk (M.T.); milan.straka@inohub.sk (M.S.); peter.bracinik@inohub.sk (P.B.)

[2] Department of Power Electrical Systems, Faculty of Electrical Engineering and Information Technology, University of Žilina, Univerzitná 8215/1, 010 26 Žilina, Slovakia

[3] Department of Mathematical Methods and Operations Research, Faculty of Management Science and Informatics, University of Žilina, Univerzitná 8215/1, 010 26 Žilina, Slovakia

[4] Department of Electric Power Engineering, Faculty of Electrical Engineering and Informatics, Technical University of Košice, 042 00 Košice, Slovakia; david.martinko@student.tuke.sk

[5] Department of International Research Projects-ERADIATE+, University of Žilina, Univerzitná 8215/1, 010 26 Žilina, Slovakia

[*] Correspondence: lubos.buzna@inohub.sk

**Abstract:** Residential chargers are going to become the standard in the near future. Their operational cycles are closely tied to users' daily routines, and the power consumption fluctuates between zero and peak levels. These types of installations are particularly challenging for the grid, especially concerning the balance of electricity production and consumption. Using battery storage in conjunction with renewable sources (e.g., photovoltaic power plants) represents a flexible solution for grid stabilization, but it is also associated with additional costs. Nowadays, grid authorities penalize a destabilization of the grid resulting from an increased imbalance between electricity generation and consumption and reward contributions to the system balance. Hence, there is a motivation for larger prosumers to make use of this mechanism to reduce their operational costs by better aligning their energy needs with the grid. This study explores the possibility of utilizing battery storage when it is not needed to fulfil its primary function of supporting charging electric vehicles, to generate some additional profit from providing a counter-imbalance. To test this idea, we develop an optimization model that maximizes the economic profit, considering system imbalance penalties/rewards, photovoltaic production, electric vehicle charging demand, and battery storage utilization. By means of computer simulation, we assess the overall operational costs while varying key installation parameters such as battery capacity and power, the installed power of photovoltaic panels and the prediction model's accuracy. We identify conditions when counter-imbalance has proven to be a viable way to reduce installation costs. These conditions include temporal distribution of charging demand, electricity prices and photovoltaic production. For the morning time window, with a suitable setting of the installation parameters, the cost reduction reaches up to 14% compared to the situation without counter-imbalance.

**Keywords:** electric vehicles; residential charging; battery storage system; photovoltaic power plant; system imbalance; counter-imbalance; speculations

## 1. Introduction

Currently, in Europe, but also in other parts of the world, significant efforts are taken to decarbonize transport and energy systems [1]. This goal is often approached through the introduction of electromobility in passenger or freight transport. Despite the great efforts of authorities, the development of electromobility is not progressing as defined by plans at the European or national levels [2]. Although the number of electric cars and charging stations is growing [2], their number is behind expectations. There are several

reasons. Some are shared across all European countries (e.g., high and growing prices of electric cars), and some are specific only to individual countries or regions (e.g., insufficient network of charging stations). The lack of charging infrastructure exists mainly because it is not profitable to operate it. Hence, addressing the problems related to the profitability of infrastructure operations is important.

Another challenge is constituted by possible negative impacts of electric vehicle (EV) charging on the power system's operation, e.g., by making it more challenging to balance the demand and energy production in the electricity system. When this is addressed properly, the charging infrastructure has the potential to be integrated into the electricity system in a network-friendly way (i.e., it can be used to deliver energy in times of electricity deficit and store the energy in times of electricity excess). Another big issue that needs to be solved is maintaining sufficient power for charging electric vehicles. The rise in EV numbers will result in an increase in the electricity demand needed in charging stations.

Altogether, it is already becoming clear that it will not be easy to fulfil the defined goals, e.g., only zero-emission cars in the EU from 2035 and a sufficient charging infrastructure or a sufficient capacity of power transmission lines, etc. [2,3]. Transmission and distribution networks will have to deal with the demand for a quick supply of large amounts of energy for fast charging of EVs, which will be required at transport hubs and along the European highway network [4]. Since the highway network is purposefully built out of the cities with good access to electrical networks, currently, the necessary amount of electricity is ensured; however, it remains a big challenge for the future. The solution for newly built charging stations appears to be the construction of local renewable energy sources (RESs) in combination with suitably sized battery energy storage systems (BESSs). This could reduce the demand for building new electrical networks with sufficient ampacity, limit the critical load (peak or total), or even support the power system operation. The combination of RESs and BESSs can also be beneficial for slow residential or public charging.

### 1.1. Literature Review

The building of microgrids equipped with local RESs is of as high importance for the development of electromobility as it is essential for the environmental benefits [5]. Since the construction of microgrids entails significant investment costs, various ways are being investigated to optimize their economic operation and thereby optimize the return on investments. One approach is to maximize the profit for the microgrid's owner through energy management while controlling the power profile of the microgrid and thus enabling the selling of energy to EVs [6]. A model for forecasting power based on real data can be used to increase the profit and maximize the usage of the power generated by the local renewable source, e.g., a photovoltaic (PV) power plant [7]. The data-driven characteristics of EV drivers can be used to evaluate their suitability to participate as a flexible demand resource [8]. The other strategy for maximizing the microgrid's profit is the management of electricity purchases and sales from and to the grid with respect to the microgrid's demand needs (e.g., EV charging demand) [9,10]. The optimization of EV battery usage and minimization of fuel consumption based on a prediction of future driving behaviour brings opportunities for energy savings [11]. Another approach is oriented toward the minimization of operational and electricity costs. This is done either through the optimization of EV charging power and actual demand in the microgrid [12–15] or the scheduling and shifting of EV charging to avoid peak prices of electricity [16–18], or the utilization of dynamic day-ahead prices in the optimization [19,20]. Coordinating the charging of electric vehicles with the day-ahead market and utilization of battery storage can reduce the overall charging cost by more than 30% [21]. Some of the proposed strategies were designed for community microgrids, e.g., [19], as community microgrids provide residential charging (charging the EVs by interconnected slow AC chargers placed directly at homes or parking places shared by the community) that can be key elements in expanding electromobility. In addition, with proper dimensioning of local energy sources and of a battery energy storage system, microgrids can contribute to the increasingly

required flexibility of the power system and, thus, through the provision of services other than just charging electric vehicles, reduce the costs of their operation, reduce the return on the initial investment or even ensure profit.

Another approach that could be exploited to optimize the operational costs is to provide support for mitigating the power system imbalance (SI) caused by the difference between generation, consumption and commercial transactions in the power system. This difference leads to the activation of regulation electricity that has to be paid for by all electricity market participants responsible for the system imbalance. Participants responsible for their own imbalance are financially penalized (rewarded) by a grid authority if they increase (decrease) the system imbalance. The behaviour of a subject leading to the decrease of system imbalance is referred to as a provision of counter-imbalance. It is worth noting that system imbalance is a different type of issue than the imbalance arising in distribution networks due to unequal loading of phases that can be, to some extent, mitigated by EV charging scheduling approaches [22]. The imbalance billing and evaluation procedures are different for each country. To gain more in rewards than to lose in penalties requires the ability to predict the system imbalance. Studies investigating the predictability of system imbalance started to appear two decades ago [23]. However, in recent years, with the rise of renewable energy generation, the characteristics of system imbalance are changing, as these energy sources are more difficult to predict, and, hence, new models are required. According to [24], system imbalance is a result of a random process caused by the combination of various factors, such as the quality of weather forecasts, output changes of power plants, and failures occurring in power systems, which indicates the complexity of predicting this quantity. Usually, the literature predicts system imbalance (SI) volume; however, it is hard to compare the absolute error of such predictions, as each country has different conditions for evaluating SI and, mainly, different SI volumes. As the imbalance settlement payment is based on the direction, i.e., the sign, of the SI, we look at the accuracies, i.e., percentages of correct guesses, of SI direction predictions. Most prediction studies estimate SI using ARIMA and machine learning models with an SI direction prediction accuracy of around 70%. Contreras [25] predicted SI in the Spanish market with a direction accuracy of around 67%. Another study [23] predicted the UK market imbalance with a 73% accuracy and Kratochvil [26] with a 70% accuracy for the Czech market. A recent study [27] improved these results using machine learning algorithms such as XGBoost and logistic regression, reaching an accuracy of around 74% for the Czech market. The highest accuracy in predicting SI directions was in [28], which reached around 90% for the Norwegian NO3 market. We see that the accuracy of SI forecast can vary, as it depends on many factors, such as the number of forecasted periods, the granularity of SI evaluations, the share of renewables in the energy mix, and possible market regulations limiting SI speculations. This supports the suggestion of [23] to measure improvements in SI forecasts with monetary savings that the predictions make possible instead of some abstract error measures.

A simple counter-imbalance strategy, which brought profit by utilizing the SI predictions, was proposed in [25]. The strategy was derived by parameterizing a cost function and by using a genetic algorithm to find parameter values that optimize costs. A similar strategy was also designed by [26], where the author showed that a profit can be achieved with binary predictions of SI. A higher profit was reached in combination with interval predictions. We identified only one study [29] performing speculations with counter-imbalance in the context of the Slovak energy market, i.e., the target market considered in this study. The authors used, as the estimate of the SI value, the SI value in the same period, but seven days ago. The battery utilization model applied SI predictions to generate some earnings on SI speculations, but the expected battery cost return was estimated at 4600 years.

As broadly illustrated by the literature review, a plethora of approaches are available to optimize the costs of residential EV charging. To the best of our knowledge, a counter-imbalance is a mechanism that has not been investigated in this context. In this paper, we identify from historical charging data candidate time windows when charging infrastructure is underutilized, and we analyse under what conditions using charging

infrastructure for speculations with counter-imbalance could be profitable. Lowering the costs of residential charging is highly relevant for several reasons. Firstly, it can help to increase the availability of charging infrastructure, make charging more convenient, and contribute to the widespread adoption of electric vehicles. Secondly, it can help reduce the need for ancillary services and investments in grid expansion. In these ways, the paper may contribute to the transition to clean energy.

### 1.2. Main Contribution and Structure of the Paper

We deliver these scientific contributions:

- We provide a data-driven feasibility study of profiting from system imbalance on residential electric vehicle charging infrastructure equipped with a battery energy storage and photovoltaic power plant.
- We present an optimization model that maximizes the profit, considering a daily EV charging plan, battery state, SI rewards, photovoltaic production, and EV charging demand while utilizing a BESS.
- We evaluate SI speculations for three different time windows, and we identify the most suitable time window and the minimal SI prediction accuracy, which can generate some economic profit and, hence, decrease the energy costs of the charging infrastructure.

The structure of the paper is as follows: Section 2 presents the data and data sources and describes the methods used in the analysis and evaluation of results. The concept of system imbalance and the imbalance market are described in brief. Section 2 also discusses the prediction models, optimization models, and the model of the considered installation on which the economic profit is evaluated. Section 3 presents the results of data analysis and computational experiments. Conclusions, limitations, and future outlooks are summarized in Section 4.

## 2. Materials and Methods

### 2.1. Datasets

#### 2.1.1. Residential Charging Dataset

The residential charging dataset contains synthetic charging power (estimations based on average charging power, session time, and actual charged energy) with hourly resolution derived from a real-world EV charging dataset containing 9757 charging sessions [30] spanning from 21 December 2018 until 31 January 2020 performed at private slow-charging infrastructure sites. The EVs charge at a maximum power of 7.2 kW. Here, we consider the data from the year 2019 as if the charging was performed in 2022.

#### 2.1.2. PV and Weather Dataset

To obtain the weather and photovoltaic (PV) data, we used the Solcast application programming interface (API) [31]. The Solcast API provides irradiation and weather forecasts, and historical data primarily customized for photovoltaics. The API provides estimates of global horizontal irradiance and temperature for selected geospatial coordinates with up to 5 min granularity.

#### 2.1.3. OKTE Dataset

We obtained the data about system imbalance and electricity prices from OKTE, which is the Slovak Short-term Electricity Market Operator. OKTE provides a public API to access data about the electricity market and system imbalance. We extracted hourly electricity prices and system imbalance data with 15 min frequency.

### 2.2. Concept of Counter-Imbalance

We consider two types of imbalance: the system imbalance and the electricity market participant's imbalance. The system imbalance is related to the mismatch between produced and consumed electricity within a certain time and area. The SI is defined as the total

amount of supplied regulation electricity to compensate for this mismatch within an imbalance settlement period, which is a 15 min time interval [32]. The SI has a positive sign if there is an excess of electricity and a negative sign otherwise. The electricity market participant's imbalance (EMPI) represents a difference between the contractually pre-agreed amount of electricity to be supplied or consumed by the participant within a certain time interval and the delivered or withdrawn amount of electricity from the grid in reality [33]. The EMPI of a subject of settlement without an off-take point is the difference between contractual electricity consumption and contractual supply according to a registered daily diagram by OKTE [34]. The EMPI has a positive sign if the subject of settlement causes an excess of electricity in the system by its behaviour (e.g., by consuming less electricity than planned) and a negative sign if the subject of settlement causes a shortage of electricity in the system by its behaviour (e.g., by supplying to the grid less electricity than planned). The difference between the SI and the EMPI is worth noting, wherein the EMPI is attributed to the electricity market participant, and the SI applies to the predefined SI settlement zone. A counter-imbalance is an EMPI of a subject that has the opposite sign compared to the SI.

OKTE penalizes or rewards market participants responsible for the imbalance. Penalties are imposed for creating an imbalance in the direction destabilizing the system, whereas rewards are given for creating an imbalance that helps the system to stabilize its frequency. Therefore, subjects responsible for imbalance are motivated to be flexible with their devices, whether they involve energy consumption or production. The process of determining SI values requires the procurement of regulation electricity, support services validation and matching trade contracts; hence, the values of the SI are determined with some delay. Hence, subjects can also be motivated to perform some SI speculations to gain rewards for counter-imbalance.

The penalty or reward in the j-th period for causing the imbalance is calculated as the amount of imbalance multiplied by the imbalance settlement price (ISP):

$$PO_j = ISP_j \times I_j \tag{1}$$

where $ISP_j$ stands for the imbalance settlement price in j-th period and $I_j$ is the imbalance of a subject at the j-th period. If $PO_j$ is negative, the subject's imbalance was in the direction of SI, and therefore, the subject pays OKTE the given amount. In the OKTE operation directives, the right side of Equation (1) is additionally multiplied by a coefficient. In practice, this coefficient is, in most cases, close or equal to one; therefore, we omitted it. According to [32], in Slovakia, $ISP_j$ is determined based on the sign of the SI. If the SI $\leq 0$, then the ISP is determined as the maximum of:

- A value of $-1$ times the positive regulation energy procurement price in the given period;
- A value of $-1.5$ times the price on the daily energy market in Slovakia in the given period. If the SI > 0, then the ISP is determined as the minimum of the following values:
- The lowest price of negative regulating electricity delivered in the given settlement period;
- A value of 1.5 times the price on the daily electricity market in Slovakia for the given period, if the price is negative;
- The price of the day-ahead electricity market in Slovakia for the given period.

Finally, if no energy was provided in the given settlement period, the ISP is equal to 100 EUR/MWh.

### 2.3. Modelled Installation

The following installation was chosen to investigate and analyse the possibility of using battery energy storage systems to provide counter-imbalance when they are not used to support electric vehicle charging. It involves a residential type of charging station, where the individual stands are placed in the garage of a residential complex. Registered electric vehicles are charged at regular intervals according to the daily routines of their users. Stands with a power of 7.2 kW connected to the AC grid are considered (AC slow charging), with a total of 70 units. Thus, the theoretical maximum charging power is 504 kW.

To support charging and relieve the power grid during peak consumption, the initial BESS size was set to 500 kWh [35], with a power of 200 kW. The battery capacity and nominal power are varied in computational experiments. The battery is managed by an energy management system (EMS) to cover the increased consumption during electric vehicle charging. We are not considering the BESS's support of ordinary household consumption. Thus, it is removed from the evaluation. The role of a photovoltaic power plant (PVPP) installation is to charge the battery and limit the consumption from the main grid. The installed power of the PVPP is a subject of experiments. The initial values are based on optimal results in [35]. The internal structure of the controlling EMS and the physical arrangement of the installation are beyond the scope of this article, and interested readers can find some examples in [35,36]. Table 1 provides a list of installation parameters for the electric vehicle charging station (EVCS), and Figure 1 depicts the principal layout diagram of the installation.

**Table 1.** Considered parameters of modelled installation of EVCS.

| Charging station | |
| --- | --- |
| Number of charging stands | 70 |
| Power of a charging stand | 7.2 kW |
| Connection type | AC |
| **BESS** | |
| Technology | Vanadium Redox Flow |
| Nominal capacity | 500/750/1000 kWh |
| Nominal power | 200/300/400 kW |
| Max discharge power | 200/300/400 kW |
| Max charge power | 200/300/400 kW |
| **Photovoltaic power plant** | |
| Installed power | 70/100/130 kWp |
| Mounting | Fixed tilt and azimuth |
| Tilt | 39.8° |
| Azimuth | 180° |
| Location | 48.093586; 17.246775 |

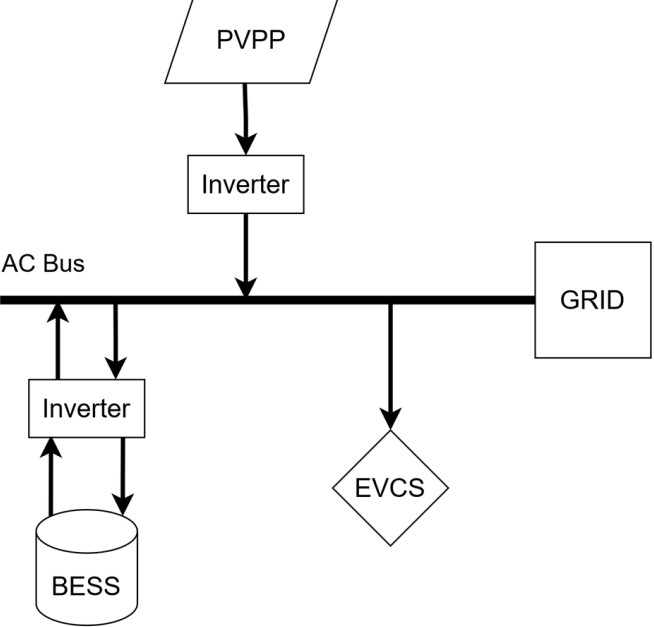

**Figure 1.** The principal layout diagram of the EVCS installation.

### 2.4. Optimization Module of the EMS

Utilizing a BESS in cooperation with an EMS is one of the most commonly used flexibility solutions. BESSs can store energy, for instance, during overproduction from a PVPP, and then enable the use of this accumulated energy during peak consumption. Such energy shifting is the primary reason for installing a BESS. However, in collaboration with a suitable EMS, a BESS can also participate in providing counter-imbalance. Opportunity for profit from counter-imbalance is expected in times when the battery is not fulfilling its primary function, such as supporting EV charging. By participating in counter-imbalance, an additional benefit is generated beyond the primary purpose of the BESS.

The EMS plans the operation of individual components of the modelled installation. The optimization module implements the chosen strategy separately in two levels, which differ in the length of the planning horizon. The layout of the planning system is depicted in Figure 2. At the D + 1 level (i.e., a day ahead of the operation), daily nominations of energy to be extracted from or supplied to the power grid are estimated together with the corresponding utilization of the BESS while considering the expected local energy consumption and expected local production from photovoltaic panels. These nominations are submitted to the organization responsible for the settlement of deviations in the operation of the electricity system. In Slovakia, this role is played by OKTE, a.s. At the H + 3 level (i.e., 3 h ahead of the operation), optimization updates the operation plan while taking into account deviation from nominations, prediction of the imbalance settlement price and updated predictions of the consumption and energy produced from PV panels. Later on, when the real consumption is known, the difference between the nominations and real consumption and the imbalance settlement price is used to determine rewards (or penalties) for counter-imbalance.

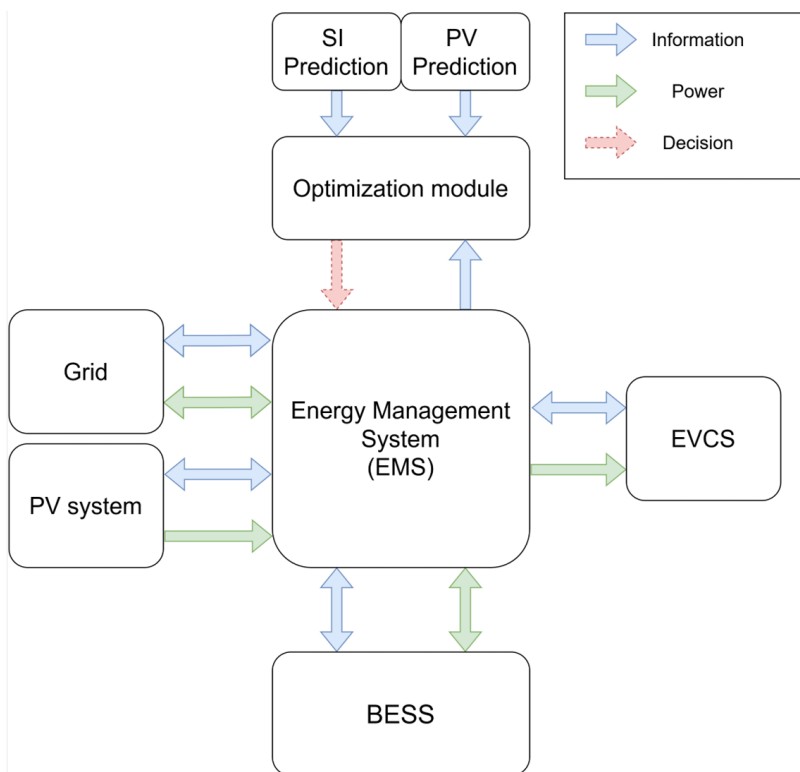

**Figure 2.** Diagram of the planning system layout.

The optimization model at the D + 1 level considers the following inputs:

$M$(big $M$)—a sufficiently large constant value (e.g., $M = 10^{10}$);
$N_{per}$—number of periods (it is assumed the considered optimization time window is discretized into this number of periods of equal duration);

$E^i_{SOLAR}$—an estimate of the maximum energy produced by PV panels in the period $i$;

$c_{SOLAR}$—costs of producing one unit of energy from PVPP (the levelized cost of energy);

$E^{BESS}_{KAP}$—the capacity of the BESS (i.e., maximum energy that can be stored);

$N_{BESS}$—number of points constituting the P vs. state of charge (SOC) diagram of the BESS (we assume that maximum power to be used to charge or discharge BESS is given by the P vs. SOC diagram. The diagram is defined by the pairs of points that share the x-coordinates, i.e., the SOC values. The y-coordinate of the first point (positive value) determines the maximum charging power for a given SOC, and the y-coordinate of the second point (negative value) determines the maximum discharging power);

$SOC_{BESS}$—an array of length $N_{BESS}$, describing the x-coordinates of the points defining SOC values in the P vs. SOC diagram;

$P^+_{BESS}$—an array of length $N_{BESS}$, describing the y-coordinates of points (positive values) defining the maximum charging power for a given SOC;

$P^-_{BESS}$—an array of length $N_{BESS}$, describing the y-coordinates of points (negative values) defining the maximum discharging power for a given SOC;

$\eta_{BESS}$—roundtrip efficiency of the BESS in storage and extraction of the energy (it is assumed that the BESS efficiency is given in the units of percent, the efficiency is the same for storage and for extraction of the energy and the roundtrip efficiency is defined as the product of these two efficiencies);

$c_{BESS}$—costs of storing/extracting one unit of energy to/from BESS (the levelized costs of storage);

$SOC^{INIT}_{BESS}$—an initial SOC value of BESS at the beginning of the first optimization period;

$SOC^{FINAL}_{BESS}$—a final SOC value of BESS at the end of the last optimization period;

$SOC_{MIN}$, $SOC_{MAX}$—the minimum and maximum SOC values;

$E^i_{CONS}$—an estimate of the energy consumed in the period $i$;

$c^i_{DEMD}$—costs per one unit of energy demanded from the power grid in the period $i$;

$RK$—reserved capacity (enumerated as the maximum energy that can be demanded from or supplied to the power grid within one period);

$c_{SUPPL}$—income received per one unit of energy supplied to the power grid.

Outputs are then characterized by the following optimization variables:

$E^i_{SOLAREXTRACT}$—energy extracted from PV panels in the period $i$;

$E^i_{BESSSTORE}$—energy stored to BESS within the period $i$;

$E^i_{BESSEXTRACT}$—energy extracted from BESS within the period $i$;

$SOC_i$—SOC of the battery at the beginning of the period $i$;

$Y^i_{BESS}$—binary variable taking a value of 1 if the energy is stored to the BESS within the period $i$, and a value of 0 otherwise;

$E^i_{DEMAND}$—energy demanded from the power grid in the period $i$;

$E^i_{SUPPLY}$—energy supplied to the power grid in the period $i$;

$Y^i_{D/S}$—binary variable taking a value of 1 if the energy is demanded from the power grid in the period $i$ and a value of 0 otherwise.

The optimization model for the D + 1 level takes the following form:

Maximize

$$
-\sum_{i=1}^{T_{per}} c_{BESS}\left(E^i_{BESSSTORE} + E^i_{BESSEXTRACT}\right) - \sum_{i=1}^{T_{per}} c_{SOLAR} E^i_{SOLAREXTRACT}
$$

$$
-\sum_{i=1}^{T_{per}} c^i_{DEMD} E^i_{DEMAND} - \sum_{i=1}^{T_{per}} c_{SUPPL} E^i_{SUPPLY}
$$

(2)

Subject to

$$
0 = E^i_{DEMAND} + E^i_{SUPPLY} - E^i_{CONS} + E^i_{SOLAREXTRACT} + E^i_{BESSEXTRACT} -
$$
$$
E^i_{BESSSTORE} \quad for \ i = 1, \ldots, T_{per}
$$

(3)

$$E^i_{SOLAREXTRACT} \leq E^i_{SOLAR} \ for \ i = 1, \dots, T_{per} \tag{4}$$

$$SOC_1 = SOC^{INIT}_{BESS} \tag{5}$$

$$SOC_{T_{per}+1} = SOC^{FINAL}_{BESS} \tag{6}$$

$$SOC_{i+1} = SOC_i + 100\sqrt{\eta_{BESS}/100}\frac{E^i_{BESSSTORE}}{E^{BESS}_{KAP}} - 100\sqrt{\eta_{BESS}/100}\frac{E^i_{BESSEXTRACT}}{E^{BESS}_{KAP}} \tag{7}$$
$$for \ i = 2, \dots, T_{per}$$

$$E^i_{BESSEXTRACT} \leq M * Y^i_{BESS} \ for \ i = 1, \dots, T_{per} \tag{8}$$

$$E^i_{BESSSTORE} \leq -M\left(1 - Y^i_{BESS}\right) \ for \ i = 1, \dots, T_{per} \tag{9}$$

$$\frac{P^-_{BESS}(j+1) - P^-_{BESS}(j)}{SOC_{BESS}(j+1) - SOC_{BESS}(j)}(SOC_i - SOC_{BESS}(j)) + P^-_{BESS}(j) \leq \frac{E^i_{BESSEXTRACT} - E^i_{BESSSTORE}}{0.25} \tag{10}$$
$$for \ i = 1, \dots, T_{per}, j = 1, \dots, N_{BESS} - 1$$

$$\frac{E^i_{BESSEXTRACT} - E^i_{BESSSTORE}}{0.25} \leq \frac{P^+_{BESS}(j+1) - P^+_{BESS}(j)}{SOC_{BESS}(j+1) - SOC_{BESS}(j)}(SOC_i - SOC_{BESS}(j+1)) + P^+_{BESS}(j) \tag{11}$$
$$for \ i = 1, \dots, T_{per}, j = 1, \dots, N_{BESS} - 1$$

$$SOC_{MIN} \leq SOC_i \leq SOC_{MAX} \ for \ i = 2, \dots, T_{per} \tag{12}$$

$$E^i_{DEMAND} - E^i_{SUPLY} \leq RK^+ \ for \ i = 1, \dots, T_{per} \tag{13}$$

$$E^i_{DEMAND} \leq MY^i_{D/S} \ for \ i = 1, \dots, T_{per} \tag{14}$$

$$E^i_{D+1} \geq -M\left(1 - Y^i_{D/S}\right) \ for \ i = 1, \dots, T_{per} \tag{15}$$

$$E^i_{SOLAREXTRACT} \geq 0, \ E^i_{BESSSTORE} \geq 0, \ E^i_{BESSEXTRACT} \geq 0, \ Y^i_{BESS} \in \{0,1\}, E^i_{DEMAND} \geq 0,$$
$$E^i_{SUPLY} \leq 0, \ Y^i_{D/S} \in \{0,1\} \ for \ i = 1, \dots, T_{per} \tag{16}$$
$$SOC_i \geq 0 \ for \ i = 1, \dots, T_{per} + 1$$

The objective function (2) expresses the overall revenues to be maximized, while the negative value corresponds to the situation when expenses exceed incomes. It is worth noting that the income from the provision of the charging service is not included. Constraints (3) ensure the conservation of energy. A set of constraints (4) limits the amount of energy extracted from the PV panels to the energy produced by the PV panels. Initial and terminal SOCs of BESS are set by constraints (5) and (6), respectively. The continuity of the SOC, while accounting for the BESS efficiency when storing and extracting the energy, is taken care of by constraints (7). As it is typically done, we consider the SOC to be defined as a percentage of the battery capacity. For this reason, stored and extracted energies are both divided by the battery storage capacity and multiplied by 100. We assume that battery roundtrip efficiency has the units of percent. Moreover, we assume that the efficiency is the same for the storage and extraction of energy and that the roundtrip efficiency is defined as the product of these two efficiencies. Therefore, the second and third terms are multiplied by the square root of the roundtrip efficiency divided by 100. A combination of constraints

(8) and (9) makes sure that energy is either stored or extracted from the BESS. Similarly, a combination of constraints (14) and (15) guarantees that energy is either demanded from or supplied to the power grid. An approximation of limitations in the rate of extraction and storage of the energy from and to the BESS (modelled by the P vs. SOC diagram) is provided by constraints (10) and (11). The SOC of the BESS is limited to the operational range by constraints (12). The demand and supply of energy from and to the power grid are limited by the reserved capacity in constraints (13). Finally, constraints (16) define the domain of all optimization variables. It is worth noting that the problem (2)–(16) belongs to mixed integer linear programming. After solving it, the nomination is given by the resulting energy demand ($E^i_{DEMAND\ SOL}$) and supply ($E^i_{SUPPLY\ SOL}$). Considering nominations and updated predictions of PV production ($E^i_{SOLAR\ UPDT}$) and energy consumption ($E^i_{CONS\ UPDT}$) as additional inputs, the optimization model at the H + 3 level is formulated with the following further inputs and optimization variables:

Additional inputs:

$c^i_{ISP}$—an estimate of the imbalance settlement price for the period *i*;
$MAX_{SPECUL}$—the maximum amount of energy limiting the deviation from nominations (as the minimum amount is considered the value $-MAX_{SPECUL}$).

Additional optimization variables:

$E^i_{DEV}$—energy representing the intentional deviation from nominations in the period *i*.

The optimization model for the H + 3 level takes the following form:
Maximize

$$
\begin{aligned}
&-\sum_{i=1}^{T_{per}} c_{BESS}\left(E^i_{BESSSTORE} + E^i_{BESSEXTRACT}\right) - \sum_{i=1}^{T_{per}} c_{SOLAR} E^i_{SOLAREXTRACT} \\
&-\sum_{i=1}^{T_{per}} c^i_{DEMD} E^i_{DEMAND} - \sum_{i=1}^{T_{per}} c_{SUPPL} E^i_{SUPPLY} + \sum_{i=1}^{T_{per}} c^i_{ISP} E^i_{DEV}
\end{aligned}
\tag{17}
$$

Subject to Constraints (2)–(15)

$$
E^i_{DEV} = E^i_{DEMAND\ SOL} + E^i_{SUPPLY\ SOL} - E^i_{DEMAND} - E^i_{SUPPLY}\ for\ i = 1, \ldots, T_{per}
\tag{18}
$$

$$
-MAX_{SPECUL} \leq E^i_{DEV} \leq MAX_{SPECUL}\ \ for\ i = 1, \ldots, T_{per}
\tag{19}
$$

$$
E^i_{DEV} \in \mathbb{R}
\tag{20}
$$

Objective (17) adds to (2) the last term that accounts for the revenue (or penalty) received by the subject responsible for the settlement of deviations from nominations. Constraints (3)–(16) are identical to the D + 1 optimization model. For every period *i*, constraints (19) define the deviation from nominations and constraints (19) limit the size of this deviation.

### 2.5. System Imbalance Prediction

We emulate SI predictions by noising the historical values of SI by adding values drawn from the normal distribution. We vary the variance of the distribution to account for different levels of prediction accuracy. This way, we explore the impact of SI prediction accuracy on the profit gained from the counter-imbalance.

### 2.6. Prediction of Photovoltaic Generation

The forecast of photovoltaic generation is an input into the optimization module. These predictions are essential for estimating the potential production capacity that will be available for charging the BESS either during the next three hours or the next day. Thus, the prediction horizon is aligned with the planning periods of optimization models presented

in Section 2.4. The prediction of achievable photovoltaic power is based on external weather forecasts. The data include air temperature, wind speed, various types of solar irradiation estimated for specific global positioning system coordinates, tilt and azimuth [31].

The standard performance calculations of a photovoltaic system are based on the solar radiation map and the potential power of the photovoltaic system per 1 kWp of installed capacity. For the regions of Slovakia, this is, on average, 1.2 kWh produced per year per 1 kWp of installed photovoltaic panels, with optimal placement of the panels with respect to angle and azimuth [37]. The power output of photovoltaic systems can be described as a relationship between weather conditions and the rated power based on standard test conditions [38]:

$$P_{PV}(t) = \eta \, PV_{PVref} * \frac{G(t)}{G_{ref}} \left(1 + \gamma \left(T_C(t) - T_{ref}\right)\right), \tag{21}$$

where $P_{PV}(t)$ is the power output of the photovoltaic system during time intervals (referred to as time granularity, e.g., 1 min, 5 min, 15 min...), $\eta$ is photovoltaic system efficiency, $G(t)$ is the solar irradiance (W/m$^2$) that falls onto a panel surface, $\gamma$ is the temperature coefficient (obtainable from datasheet) accounting for the change in the power when the panel temperature $Tc(t)$ deviates from the reference temperature. Subscript *ref* refers to panel standard testing conditions where $G_{ref}$ = 1000 W/m$^2$ and $T_{ref}$ = 25 °C. $PV_{PVref}$ is the maximum panel power output at reference conditions.

The Koehl model [39] for calculating the heating of the panels as a function of ambient temperature, wind speed and incident radiation was selected. It utilizes meteorological data and empiric constants to estimate panel temperature, $Tc(t)$. Installation with a fixed position, tilt and the azimuth of panels were considered [40]. Thus, $G(t)$ represents irradiance on a tilted surface, obtainable either directly from a data source (e.g., Solcast API) or as the composition of direct, diffuse and reflected irradiance.

### 2.7. Evaluation of Results

The evaluation of results is based on the simulation of a charging and discharging battery according to outputs from the optimization model, with respect to currently available energy. The main purpose of the simulation is to validate optimization results in near-real operation conditions. The optimization model results, based on predictions, consider the amount of energy that may not be available under real conditions. To calculate the currently available energy, the simulation firstly covers the entire consumption of the charging station by the production from the PV power plant. Then, the extra PV energy is stored in the battery. Otherwise, when the PVPP production is insufficient, the lacking energy is recovered from the battery.

The costs associated with battery usage and maintenance are calculated based on economic parameters. If the battery is 100% charged, the excess energy from the PVPP is sold to the grid (according to the current price); on the other hand, if the battery is at a 0% SOC, the consumption must be covered by supply from the grid. The BESS is charged exclusively from the PVPP because charging the battery from the grid during covering consumption creates additional costs. The prices for electrical energy are evaluated, including distribution charges for consumption and supply to the grid. The energy flow between individual units of the installation is shown in Figure 3. The BESS losses are included in Equation (7), PV losses are considered in Equation (21) and transmission losses are neglected.

The first step of evaluation is the calculation of the baseline state without considering a counter-imbalance, against which all other scenarios are subsequently compared. An important parameter obtained from the baseline state of the installation, apart from the total costs, is the SOC of the battery at the beginning and end of the counter-imbalance provision time window. These values are used by the optimization module, which must consider the battery's initial SOC state at the beginning of the counter-imbalance provision window but, more importantly, calculates the required SOC level the battery

must reach by the end of the window to ensure optimal support for charging throughout the rest of the day. Additionally, there is an effort to maximize the profit from counter-imbalance and minimize additional costs. Economic evaluation, cost and profit calculations are carried out only within the chosen time window; the rest of the day remains the same for all scenarios and is therefore not economically evaluated. The compared scenarios and studied variations in the approach to providing counter-imbalance are described in more detail in the following section.

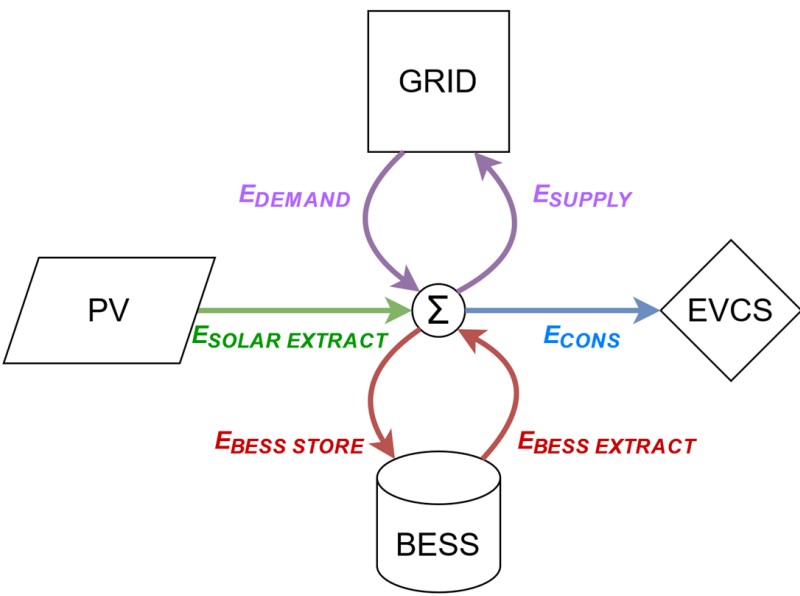

**Figure 3.** The energy flow diagram of the installation.

## 3. Results

This section starts with data analyses of candidate time windows for SI speculation. Afterwards, the setup of planned experiments is listed. Finally, the numerical and graphical results of the experiments are presented.

### 3.1. Data Analyses

Here, we analyse time windows selected for speculating with counter-imbalance based on charging power, PV power, and SI prices. To ensure that we performed analyses on different data than the computational experiments, we divided the EV charging dataset into two parts. The first part precedes, and the second follows, 1 May 2019. We looked for periods when the charging stations were idle, i.e., the BESS had sufficient power to provide counter-imbalance. Figure 4 depicts the total charged energy at a given hour of the day for the analysed period. The minimum values were observed in the early morning from 5:00 to 8:00, which we selected as the first time window. Also, at this time, the prices on the day-ahead market (DAM), displayed in Figure 5, reached the morning peak. These prices are important as they are input to the SI price calculation. The second time window was selected as from 10:00 to 13:00 due to the combination of peaking PV power, depicted in Figure 6, and low EV charging demand. To benefit from the high SI prices while the battery could be charged from the PV power, we selected the last window as from 16:00 to 19:00. It is necessary to point out that the selected time windows were evaluated independently of each other, and, thus, the optimization and subsequent SI speculations were conducted in each time window as separate experiments.

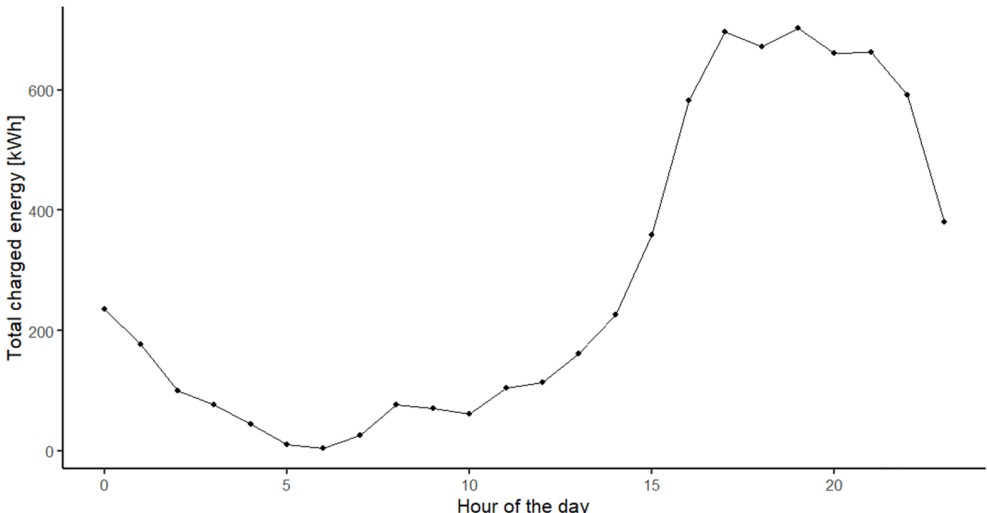

**Figure 4.** Hourly distribution of the energy charged at the charging stations.

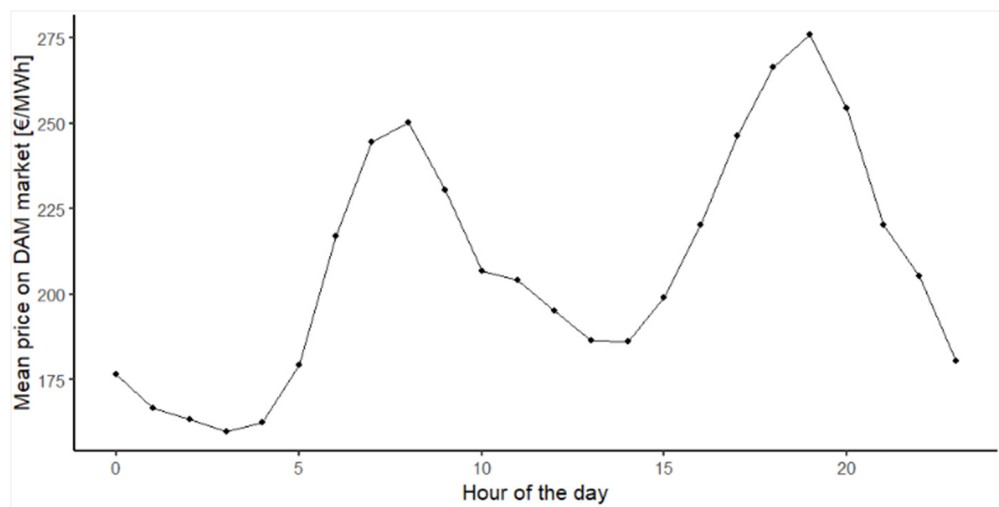

**Figure 5.** Mean electricity prices at DAM market in Slovakia.

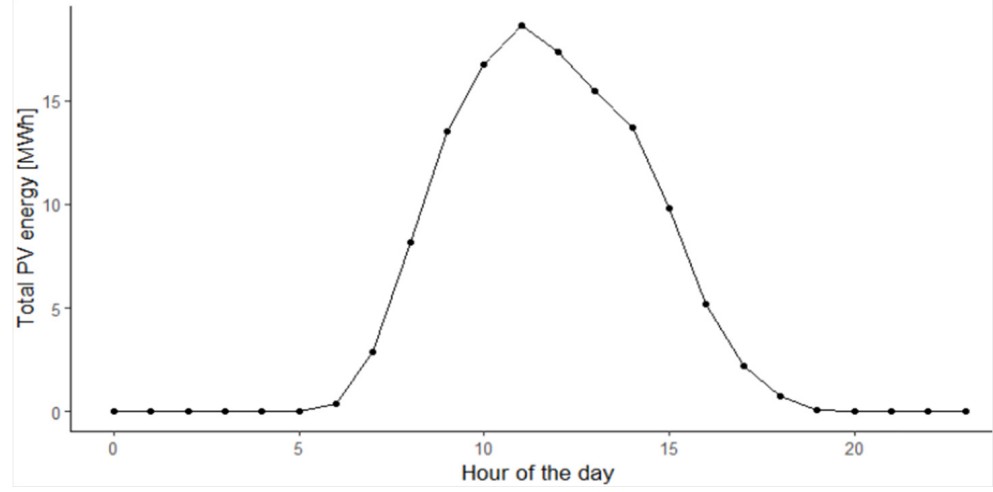

**Figure 6.** Total PV power produced in Slovakia at maximum power: 70 kWp.

### 3.2. Setting of Parameters

To investigate the profitability of providing counter-imbalance, simulation experiments were conducted for a range of input parameters to inspect the influence of individual parameters on the total energy cost of the system. We performed experiments with all possible combinations of parameters, displayed in Table 2.

**Table 2.** Setting of parameters.

| Parameter | Values | | | |
|---|---|---|---|---|
| Time window | 5:00–8:00 | 10:00–13:00 | 16:00–19:00 | |
| BESS capacity | 500 kWh | 750 kWh | 1000 kWh | |
| BESS power | 200 kW | 300 kW | 400 kW | |
| PVPP power | 70 kWp | 100 kWp | 130 kWp | |
| $MAX_{SPECUL}$ | 12.5 kWh | 25 kWh | 50 kWh | 100 kWh |
| SI prediction accuracy | 70% | 80% | 90% | 100% (Oracle) |

The initial value of the BESS capacity, 500 kWh, represents the minimum capacity to reliably cover the average daily energy demand of the modelled installation. Higher capacities are capable of covering more demand, enabling more effective power shifting and providing higher potential for profiting from counter-imbalance. The disadvantage of a higher capacity is the increased cost of the BESS.

The power of a Vanadium Redox Flow BESS (VRFB) is scalable independently from its capacity. The presented values are based on datasheets of real BESSs [41]. Moreover, the maximal charge and discharge power can be up to double the nominal power with this battery technology. Limitations are represented by charge and discharge characteristics, shown as the dependence of power on the SoC in Figure 7.

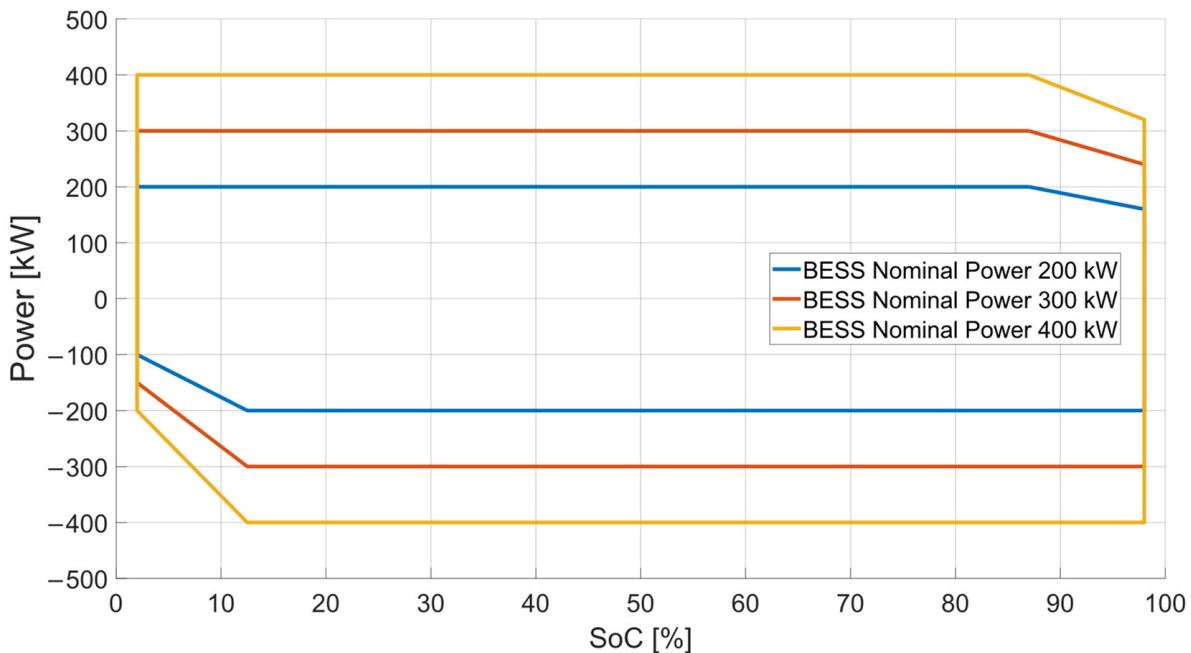

**Figure 7.** SoC vs. power characteristic of the considered VRFB.

A BESS with more power can manipulate a larger amount of energy and, hence, can bring higher profits, mainly through economically advantageous periods. It is not advisable to use a BESS with excessively large power. The main reasons are economic, including investment costs and operating and maintenance expenses. Additionally, it is necessary to consider the physical and spatial dimensions of the installation and the energy infrastructure at the installation site. Local legislative restrictions are also one of the limiting

factors. For relatively short lengths of investigated periods and specific VRFB technology, the degradation of the VRFB is often neglected in operation simulations and economic modelling, as in our case. However, the estimated number of cycles was computed to indicate an impact on battery life.

The considered range of installed PVPP power was inspired by articles focused on the optimization of EVCS parameters [35,36]. A larger installed power means higher revenue from selling energy to the grid, faster and cheaper charging of the battery to cover peak or nighttime consumption and a reduced impact of lower production from the PVPP (e.g., in winter months).

The battery capacity and the installed power of the PVPP are the main parameters that determine the costs of using these devices. The levelized cost of storage (denoted as $c_{BESS}$ in the optimization model) is a commonly used economic indicator that summarizes the costs of using a BESS, depending on initial investments, operation and maintenance costs, expected lifetime and estimated number of cycles per year. This value is calculated with regard to the primary purpose of the BESS. For a PVPP, the equivalent indicator is the levelized cost of energy (denoted as $c_{SOLAR}$—in the optimization model). Estimated values of these parameters are listed in Table 3 and are used for cost evaluation in experiments.

**Table 3.** The setting of values of economic parameters.

| BESS Capacity | 500 kWh | 750 kWh | 1000 kWh |
|---|---|---|---|
| $c_{BESS}$ | 0.252 EUR/kWh | 0.371 EUR/kWh | 0.489 EUR/kWh |
| Installed PVPP power | 70 kWp | 100 kWp | 130 kWp |
| $c_{SOLAR}$ | 0.0705 EUR/kWh | 0.0705 EUR/kWh | 0.0705 EUR/kWh |

The optimization module restricts the size of SI speculations by limiting the minimum ($-MAX_{SPECUL}$) and maximum power ($MAX_{SPECUL}$) that can be used to provide counter-imbalance. A low SI limit represents a cautious approach to counter-imbalance, with lower financial risk. A high SI limit has the potential for higher earnings, but also, it may result in higher penalties for incorrect counter-imbalance. This parameter also directly determines the battery power allocated for providing counter-imbalance. Suppose this limit is greater than the battery's power. In that case, the system can use the entire battery during the counter-imbalance window, leaving no room for charging electric vehicles. However, this is not an issue during periods of minimal power consumption.

A more powerful prediction model provides more accurate predictions for the optimization module. This increases the potential earnings from counter-imbalance while losses from penalties are diminished. Additionally, the costs of using the battery and drawing power from the grid decrease due to more efficient utilization of the energy flow between installation components. To investigate the dependence between the profit from counter-imbalance and the required level of the precision of SI predictions, we vary the precision in the range from 70% to 100% (see Section 2.5). The value 100% corresponds to a prediction with zero error (Oracle model), and thus, in such a case, we use real data instead of predictions in the optimization module to demonstrate the maximum potential of counter-imbalance.

### 3.3. Results for Scenario without Providing Counter-Imbalance—Baseline

In the first series of experiments, we investigated the baseline scenario without providing counter-imbalance. We evaluated the costs of using a BESS, the cost of a PVPP, estimated earnings from sold energy based on DAM prices and estimated costs of purchased electricity from the grid based on DAM prices. Further, we evaluated the total cost of acquiring energy for charging electric vehicles for every combination of parameters. The earnings from the provision of the EV charging service are beyond the scope of this article and are not considered in the evaluations.

The baseline experiments confirmed that the installation generates different total costs in various time windows (see Figure 8). The lowest costs in most of the experiments

occurred in the midday window from 10:00–13:00 due to it evidencing the highest PVPP production and low EV consumption. The negative value of costs represents profit from sales of the surplus energy to the grid. In the early evening window from 16:00–19:00, the highest value of cost was observed in the majority of experiments. During this period, EV consumption reaches its maximum, and PVPP generation gradually decreases. The BESS is discharged to cover the consumption, and the remaining energy needs are drawn from the grid. Moreover, the dependence of total costs on the installation parameter values was the most prominent in this time window. On the other hand, the costs in the morning window, 5:00–8:00, were in the middle of the spectrum, and dependencies on the installation parameters were stable due to it evidencing the lowest EV consumption and almost negligible PVPP production. The battery is practically not actively engaged during this period, and interaction with the grid is also minimal.

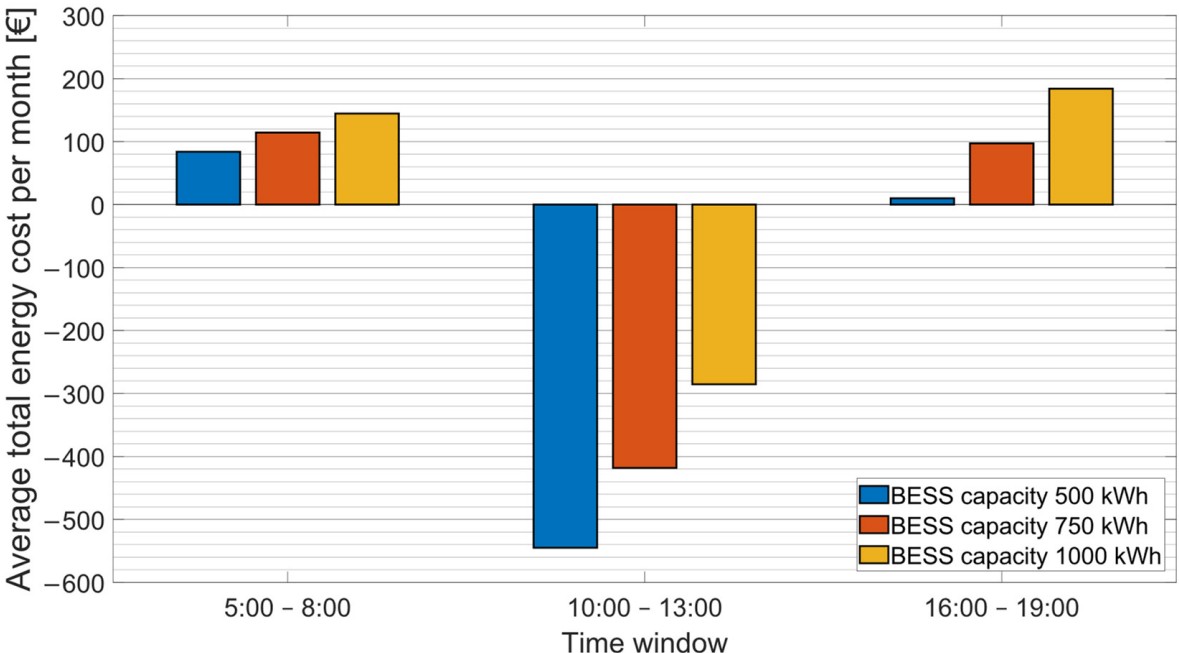

**Figure 8.** Average of total energy cost per month, comparison of BESS capacity.

The average total monthly costs of acquiring energy for EV charging as a function of the BESS capacity can be seen in Figure 8. Considering the BESS capacity as given, we report results for the best combination of other parameter values. For the morning window, the PVPP installed power was 70 kWp, and for the midday and early evening window, it was 130 kWp. In all the baseline experiments, the BESS power parameter had no impact at all.

In Figure 8, the impact of the battery capacity size on the total energy cost is evident. Naturally, a battery with a larger capacity is more expensive, and its usage generates more costs if the volume of processed energy remains the same. Furthermore, it was found that changes in the battery's nominal power do not affect the overall system costs. The reason is that even the smallest selected power rating of 200 kW is sufficient to absorb the difference between simulated PVPP production and EVCS consumption.

Another parameter that significantly impacted the total energy cost of the baseline experiments was the PVPP installed power (see Figure 9). It is evident that during the midday and early evening time window, a higher installed PVPP power significantly decreases the total energy costs. The generated energy fully charged the battery, and the rest was sold to the network for a profit, especially during the period of maximum production within the midday window. However, as the morning window behaves differently, using a PVPP with smaller power is more effective during this period. The reason is the depleted battery, which has to be charged with the energy generated by the PVPP, leaving a negligible

amount of energy to be sold to the grid. Even with the highest value of considered PVPP power, the system was not able to fully charge the battery due to the low irradiance during this time window.

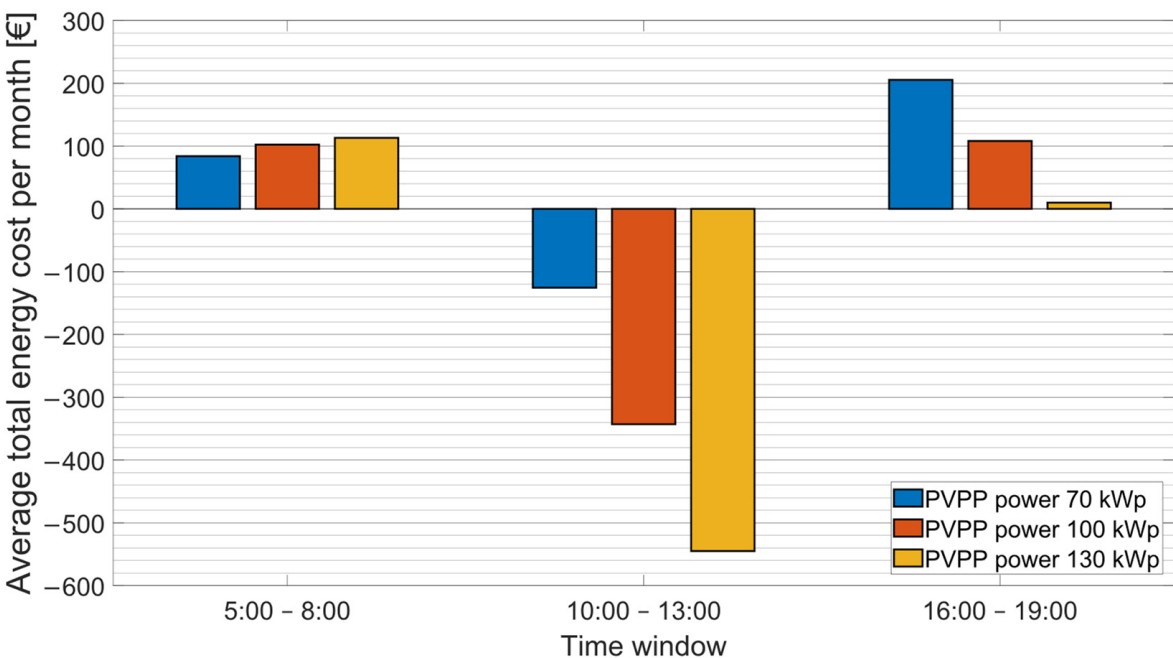

**Figure 9.** Comparison of the average total energy costs per month for different PVPP installed powers.

The comparison of the average total costs when varying the PVPP power is shown in Figure 9. In all time windows, the lowest costs were achieved with a BESS capacity set to 500 kWh, i.e., the smallest capacity of the battery.

Table 4 summarizes the values of parameters leading to the minimum total costs for each considered time window and the baseline scenario. A more detailed breakdown of individual energy cost components is also presented. We omitted the BESS power parameter, as it did not impact the cost. The total cost of energy can be interpreted as the cost of procuring energy for the purpose of charging electric vehicles. All costs and earnings in the following tables are monthly averages.

**Table 4.** Results obtained for the best combination of installation parameters, leading to the smallest costs.

| Parameter | Time Window | | |
|---|---|---|---|
| | **5:00–8:00** | **10:00–13:00** | **16:00–19:00** |
| BESS capacity | 500 kWh | 500 kWh | 500 kWh |
| PV power | 70 kWp | 130 kWp | 130 kWp |
| Number of BESS cycles | 0.253 | 1.063 | 0.727 |
| The cost of using BESS | EUR 64 | EUR 268 | EUR 183 |
| The cost of using PVPP | EUR 20 | EUR 412 | EUR 125 |
| The cost of grid electricity | EUR 8 | EUR 2 | EUR 73 |
| Earnings—sales to grid | EUR 7 | EUR 1228 | EUR 371 |
| Total energy cost per month | EUR 85 | EUR −546 | EUR 10 |

Through the analysis of individual cost components, it becomes evident that the most significant impact on total costs can be attributed to grid sales. This component is directly dependent on the installed power of the PVPP. The operational costs increase with growing installed power, but the profit from selling energy to the grid surpasses these expenses. The cost of purchasing energy from the grid reflects consumption levels. During the early

evening window, when the PVPP, in conjunction with stored energy in the battery, does not cover the entire consumption, purchases of energy from the grid are necessary. BESS costs peak during the midday window as it represents the period when the battery is charged, and the highest amounts of energy are manipulated, which is confirmed by the estimated cycle count.

### 3.4. Results for Scenario with Counter-Imbalance

The second set of experiments involved evaluating the benefits of providing counter-imbalance using predictions of the system imbalance, photovoltaic production, and charging station consumption. These predictions were not accurate, so the recommended distribution of power flow delivered by the optimization module may differ from actual power flows, potentially turning the expected gains from counter-imbalance into penalties for not adhering to the reported nomination.

By evaluating all combinations of input parameters, it was revealed that the prediction accuracy fundamentally impacts the success of providing counter-imbalance. Regardless of the values of other parameters, a reduction in the overall costs was achieved only during the morning time window and in experiments where the prediction accuracy was at least 90%. Profits from providing counter-imbalance can be achieved even with less accurate forecasts, but they do not reach sufficient levels to achieve the desired effect of reducing the overall energy costs (e.g., to overweight the costs of storing or extracting the energy from the BESS). The profitability threshold for the time window from 5:00 to 8:00 is between 80% and 90%. In other time windows, for the prediction accuracy of less than 100%, the counter-imbalance did not bring the expected cost reduction.

In Figure 10, the influence of prediction accuracy on the overall costs is analysed for the best combinations of parameters, according to Table 5. The dotted line marks the total energy cost for the baseline scenario. Thus, a desired result for providing counter-imbalance should lead to costs that lie under the dotted line.

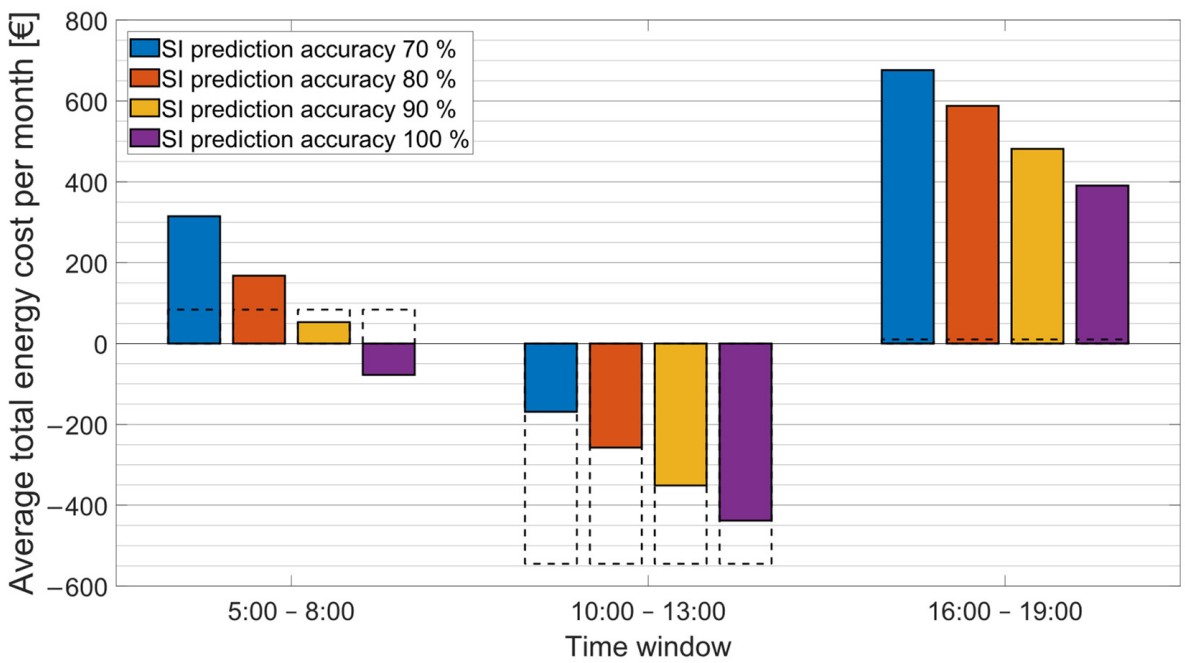

**Figure 10.** Comparison of average total energy costs per month for different SI prediction accuracies. The dotted lines denote the total energy cost for the baseline scenario.

It should be noted that the cost reduction did not occur even when using perfectly accurate SI predictions in the midday and early evening time windows. The reason is that other sources of uncertainty (prediction accuracy of PVPP production and EV charging demand) cause a difference between the plan suggested by the optimization model and its

realization that result in increased average total costs. The speculation limit ($MAX_{SPECUL}$) is also a significant aspect contributing to SI speculation. A larger speculation limit means a greater amount of energy for counter-imbalance, which could lead to a higher profit. However, experiments have shown that it is generally more advantageous to minimize risk and set a low SI speculation limit because penalties in the case of widening the limit increase the costs disproportionally. Figure 11 presents a cost comparison for the considered SI speculation limit values with the accuracy of SI predictions set to 90% and parameters according to Table 5.

**Table 5.** Results obtained for the combination of parameters leading to the smallest costs in experiments with a counter-imbalance provision.

| Parameter | Time Window | | |
|---|---|---|---|
| | **5:00–8:00** | **10:00–13:00** | **16:00–19:00** |
| BESS capacity | 500 kWh | 500 kWh | 500 kWh |
| BESS power | 400 kW | 300 kW | 200 kW |
| PV power | 70 kWp | 130 kWp | 130 kWp |
| Max SI speculation | 25 kWh | 12.5 kWh | 12.5 kWh |
| Number of BESS cycles | 1.772 | 1.461 | 1.264 |
| The cost of using BESS | EUR 447 | EUR 368 | EUR 319 |
| The cost of using PVPP | EUR 20 | EUR 412 | EUR 125 |
| The cost of grid electricity | EUR 388 | EUR 112 | EUR 319 |
| Penalties imbalance | EUR 115 | EUR 236 | EUR 508 |
| Earnings—sales to grid | EUR 375 | EUR 1302 | EUR 627 |
| Earnings—imbalance | EUR 542 | EUR 178 | EUR 161 |
| Total energy cost per month | EUR 53 | EUR −352 | EUR 483 |

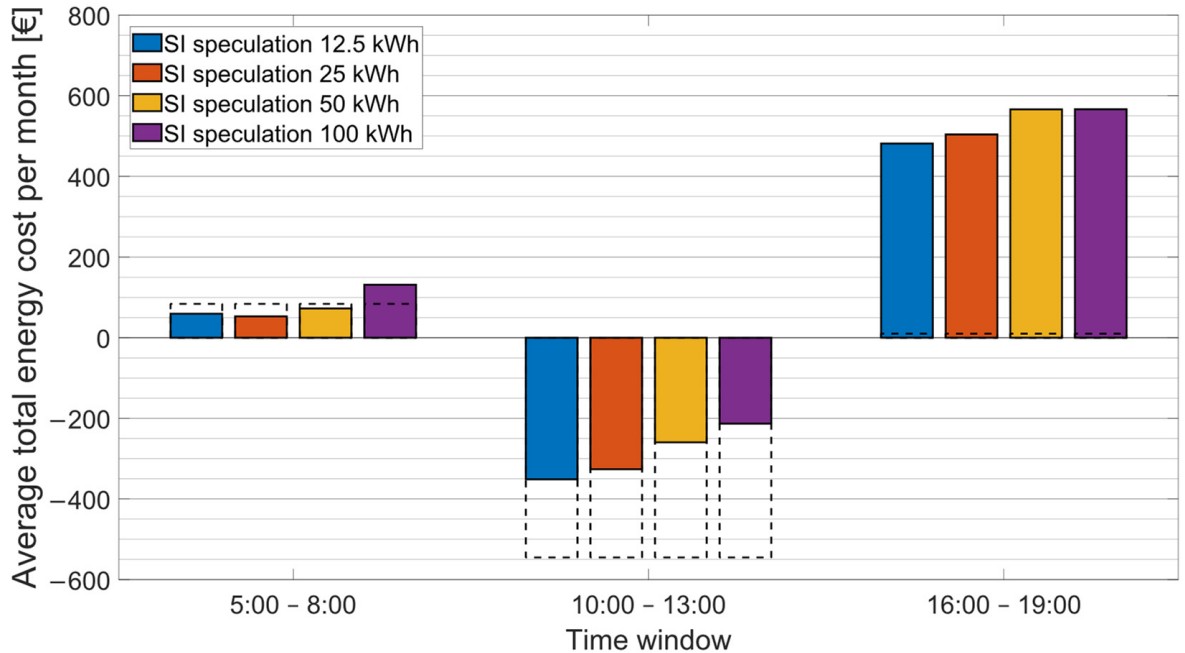

**Figure 11.** Comparison of average total energy costs per month for different SI speculation limits. The dotted lines denote the total energy cost for the baseline scenario.

The only profitable time window for providing counter-imbalance was from 5:00 to 8:00. In this window, SI speculation achieved the desired reduction in total energy costs. The reasons for this are favourable DAM prices and very good predictability of charging station consumption. The counter-imbalance did not reduce overall energy costs in the other time windows. Increasing the PVPP installed power proportionally increases

direct sales to the grid, similarly to the baseline scenario. Moreover, the optimization module could not fully utilize this additional energy for counter-imbalance due to the speculation limit. Thus, the installed power of the PVPP does not have a significant impact on providing counter-imbalance. Increasing the battery capacity has a more significant impact on the total energy cost from providing counter-imbalance, as is shown in Figure 12. This figure compares different battery capacities for parameters shown in Table 5 and has an SI prediction accuracy set to 90%. It has been found that it is more advantageous to use a smaller battery despite cumulating more cycles, rather than a larger battery with fewer cycles cumulated, due to battery costs.

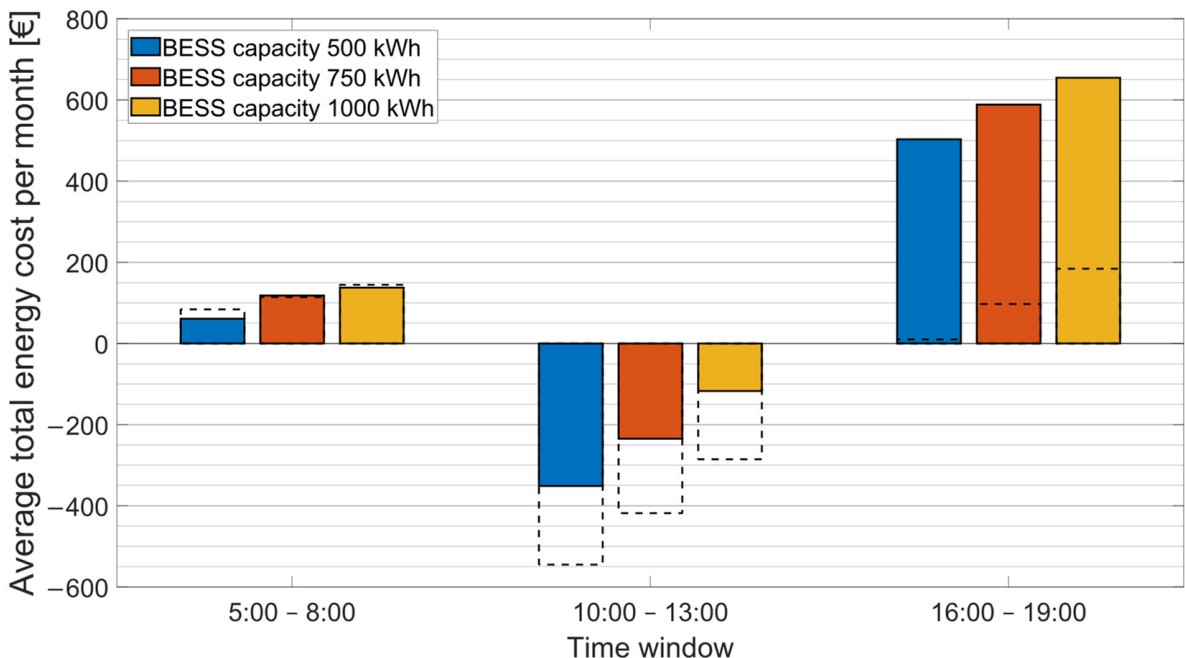

**Figure 12.** Comparison of average total energy costs per month for different BESS capacities. The dotted lines denote the total energy cost for the baseline scenario.

From Figure 12, it is evident that the impact of increasing BESS capacity is similar to the baseline scenario. In addition, SI speculations exploit the battery to manipulate energy from and to the grid according to counter-imbalance, which naturally generates some costs. For the morning time window, we can see that the combination of a larger battery with SI speculation has the potential to even the baseline energy costs, but generally, an overrating of BESS capacity is ineffective.

Table 5 summarises experimental results with the provision of counter-imbalance with an SI prediction accuracy set to 90% and presents the best combinations of installation parameter values with achieved costs and profits. The parameter of BESS power was different in every time window, corresponding to a decline in SI speculation effectiveness in the midday and early evening time windows. Each time window was evaluated separately and independently, and the total energy costs are directly comparable with the baseline scenario results in Table 4. In the comparison with the baseline scenario, it is worth noticing the increased cost of using a BESS and the increased costs of purchased grid electricity. These costs are incurred by the energy manipulations necessary for SI speculation.

Finally, we evaluate the revenue, representing a difference between earnings and penalties resulting from providing counter-imbalance, with an SI prediction accuracy of 90%, in Figure 13. The revenue from providing counter-imbalance is mostly positive; however, it may not be sufficient to reduce the total cost of energy. From Figure 13, it is visible that with an increase in the SI speculation limit, the SI revenue also increases. However, excessive increases in the speculation limit eventually do not result in higher

profits because other constraints are more restrictive, such as the maximum battery power. The SI speculation limit directly determines how much battery power and capacity needs to be allocated to provide the counter-imbalance and what level of risk we are willing to accept. In summary, the main parameters influencing the profit from counter-imbalance are the SI speculation limit and prediction accuracy.

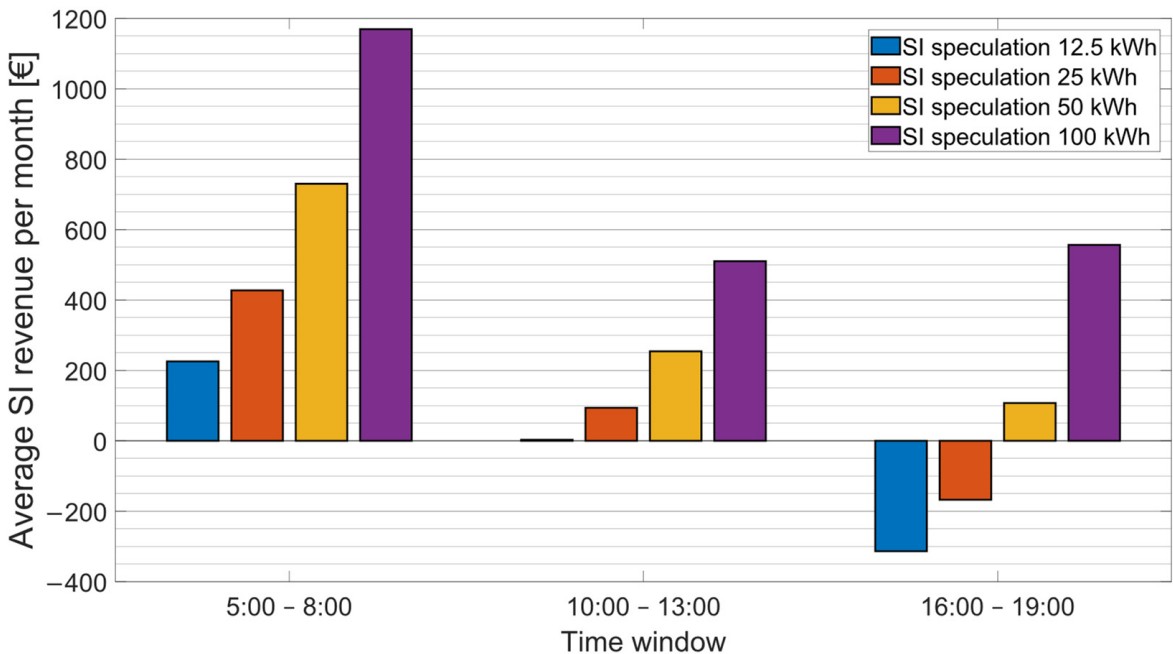

**Figure 13.** Comparison of average monthly SI speculation revenues for different SI speculation limits.

Table 6 summarizes installation parameters for the best scenarios from the point of view of SI revenues, with the SI prediction accuracy set to 90%. The applicability of the principle of high risk—high reward is apparent from the comparison of earnings and penalties. Still, SI revenue is insufficient to reduce the total cost compared to the baseline scenario with the same values of parameters (see Table 4).

**Table 6.** Results obtained for the combinations of parameters leading to the highest SI revenues.

| Parameter | Time Window | | |
|---|---|---|---|
| | **5:00–8:00** | **10:00–13:00** | **16:00–19:00** |
| BESS capacity | 500 kWh | 500 kWh | 500 kWh |
| BESS power | 400 kW | 400 kW | 400 kW |
| PV power | 70 kWp | 70 kWp | 70 kWp |
| Max SI speculation | 100 kWh | 100 kWh | 100 kWh |
| Number of BESS cycles | 4.799 | 3.849 | 4.964 |
| The cost of using BESS | EUR 1210 | EUR 970 | EUR 1252 |
| The cost of using PVPP | EUR 20 | EUR 222 | EUR 67 |
| The cost of grid electricity | EUR 1078 | EUR 730 | EUR 1484 |
| Earnings—sales to grid | EUR 1007 | EUR 1188 | EUR 1309 |
| Earnings—imbalance | EUR 1505 | EUR 963 | EUR 1484 |
| Penalties imbalance | EUR 336 | EUR 453 | EUR 928 |
| SI revenue per month | EUR 1169 | EUR 510 | EUR 556 |

Table 7 summarizes the results from counter-imbalance experiments while using perfectly accurate SI predictions (Oracle). These results can be used as a reference case to assess the impact of SI prediction accuracy. Compared to Table 6, the most significant difference is in penalties for the wrong direction of the provided counter-imbalance. The

relation between SI prediction accuracy and the decrease in penalties is nonlinear, caused by changing imbalance prices throughout investigated periods. Perfectly accurate SI predictions minimize penalties, while earnings remain roughly the same due to an identical amount of available energy, as the installation parameters are also identical. Yet, the optimization with SI Oracle predictions can use this available energy more effectively. Despite this, penalties occurred in every time window, caused by other uncertainties. In addition, the combination with a perfectly accurate prediction illustrates the maximum potential for cost reduction by providing counter-imbalance.

**Table 7.** Comparison of average monthly SI revenues for Oracle predictions.

| Parameter | Time Window | | |
|---|---|---|---|
| | **5:00–8:00** | **10:00–13:00** | **16:00–19:00** |
| BESS capacity | 500 kWh | 500 kWh | 500 kWh |
| BESS power | 400 kW | 400 kW | 400 kW |
| PV power | 70 kWp | 70 kWp | 70 kWp |
| Max SI speculation | 100 kWh | 100 kWh | 100 kWh |
| Number of BESS cycles | 4.393 | 3.520 | 4.064 |
| The cost of using BESS | EUR 1108 | EUR 887 | EUR 1025 |
| The cost of using PVPP | EUR 20 | EUR 222 | EUR 67 |
| The cost of grid electricity | EUR 967 | EUR 653 | EUR 1180 |
| Earnings—sales to grid | EUR 912 | EUR 1121 | EUR 1094 |
| Earnings—imbalance | EUR 1515 | EUR 1070 | EUR 1440 |
| Penalties imbalance | EUR 26 | EUR 106 | EUR 460 |
| SI revenue per month | EUR 1489 | EUR 964 | EUR 980 |

## 4. Discussion

A direct comparative analysis of the studied scenarios can be facilitated by Tables 4 and 5, presenting the best results obtained for the baseline scenario and the scenario with a counter-imbalance provision. SI speculations increase the energy exchange with the grid, as evident from the comparison of electricity sales and purchases from the grid in Tables 4 and 5. During the midday window, grid sales dominate in both tables and constitute the main component of the total costs due to the peak PVPP production. The comparison of the morning windows demonstrates a significant increase in grid interaction, battery costs, and the corresponding cycle count. This is a result of the counter-imbalance provision, which ultimately led to the energy cost reduction, an effect that was not achieved in other time windows.

Tables 6 and 7 support the comparative analysis of the revenues (a difference between earnings and penalties) derived from SI speculations. A high speculation limit in combination with a high battery capacity is required for a higher speculation profit. This is absent in Tables 4 and 5 because SI speculations do not reduce the overall costs, despite significant counter-imbalance revenue. A higher SI speculation limit without sufficiently accurate predictions cannot generate enough profit to cover created costs. Table 7 represents the maximum profit potential from counter-imbalance, using 100% accurate SI predictions. The main difference between Tables 6 and 7 is in the imbalance penalties, which are significantly increased if SI predictions are inaccurate and result in lower overall revenues.

This study includes several sources of uncertainty stemming from predictions of input data for the optimization model, namely, predictions of SI, EVCS consumption, and PVPP production. As previously stated in Section 3.4, the results were mostly affected by uncertainties related to SI predictions. Generally, predicting SI is a challenging problem that requires advanced prediction tools. Correctly determining the direction of SI is fundamental to profit because providing counter-imbalance in the opposite direction results in penalties, which was found to be a reason for low earnings. We established an approximate prediction accuracy threshold, determining when providing counter-imbalance could be profitable for a considered application case. A more sophisticated SI prediction model could open

possibilities for more complex speculation strategies. For example, a prediction model indicating prediction reliability could be used to determine periods when SI speculations are less risky. A missing risk assessment in SI speculations is one of our study's limitations, as SI speculations were carried out in each period of the selected time windows, even in cases of high penalty risk or unfavourable DAM prices.

Uncertainties in EV charging demand predictions directly impact the available energy for SI speculations. The assumed reserved energy may not be actually available due to a higher number of connected vehicles, resulting in less energy for counter-imbalance and lost profit. Conversely, when EV charging demand is overestimated, profit from counter-imbalance can be reduced, e.g., by selling the excess energy to the grid. In the case of very inaccurate EV charging demand estimation, it may even happen that no energy is available for counter-imbalance, and the energy deficit is covered by purchasing from the grid. Improvements in estimations can be achieved by deeper analysis and comprehension of consumer behaviour.

The prediction of PVPP output is directly linked to predicted meteorological data, each having its own level of uncertainty. The most significant impact on the resulting predicted PVPP output comes from uncertainties in irradiance forecasts. Other input meteorological data, such as ambient temperature and wind speed, have only a minor influence. An inaccurate estimate of PVPP production similarly impacts the assumed amount of available energy as EV charging demand misestimation.

The mentioned sources of uncertainty in the input data are not subjected to risk evaluation, nor does the used optimization model consider the uncertainties; therefore, the proposed methodology represents a deterministic scheduling algorithm. However, the impact of some uncertainties is captured in results, e.g., in the total energy costs or in the profit from counter-imbalance. On one hand, we admit that the aspect of uncertainty needs to be addressed in a more complex way, e.g., using more sophisticated optimization models such as the uncertainty set-based robust optimization [42]. On the other hand, when deploying more complex models and associated optimization methods in practice, increased computational complexity must be taken into account. We assume that final decisions may be taken by human operators based on the recommendations provided by optimization results. Thus, optimization results may need to be delivered multiple times within each 15 min period.

As another limitation of the presented study, the evaluation procedure of the overall system costs can be outlined, as we experienced a lack of data on EV charging prices. Without this information, it was not possible to assess the direct revenues from the primary EVCS business. It can be assumed that, in practice, revenues must fully cover the operational costs of the installation. Therefore, additional income from SI speculation services could, in practice, lead to a reduction in EV charging prices. With the evolving technology of BESS and PVPP installations, a further decrease in operation costs can be expected, contributing to the viability of SI speculations.

Finally, there remains the question of negative impacts of the proposed methodology. The effects of devices connected to the grid through inverters, such as BESSs or some types of EV charging stations, are generally known, and widespread deployment of these devices on a large scale undoubtedly negatively affects the grid. However, these impacts need to be addressed prior to their use, in the design and installation process of the devices, taking into account local conditions in the grid. An analysis of negative impacts on system stability is beyond the focus of our article. The primary purpose of our proposed methodology is to reduce operating costs for residential EV charging stations supported by BESSs while adding the benefit of supporting system stability rather than solving the system stability problem itself. Since our methodology works with the counter-imbalance concept and is more associated with intervention in the control systems of BESSs than in the hardware itself, we believe that widespread deployment would positively impact grid stability without worsening existing negatives or introducing new adverse effects.

However, providing counter-imbalance negatively impacts the lifespans of BESSs, as is evident from the comparison of the predicted number of cycles. Reduced lifespan and increased wear of BESSs are natural effects of intensified energy manipulations. This phenomenon was considered in our analysis, as in some experiments, the increased costs due to intensive manipulation exceeded the profit from SI speculation. Ensuring the economic efficiency of providing counter-imbalance is an optimization task that, in our case, requires the identification of suitable parameters and time windows for speculation. Under these conditions, the profit from counter-imbalance compensates for the negative impact of reduced lifespan and increased wear of a BESS.

## 5. Conclusions

By means of computer simulation, we studied the economic feasibility of utilizing a residential electric vehicle charging station with a battery energy storage system and photovoltaic powerplant for speculations with system imbalance. To derive insights about the suitable configuration of the system, we varied the parameter values in computation experiments. The provision of the counter-imbalance can reduce installation costs by using a BESS at times when it is not performing its primary function. By performing data analysis of EV charging power demand, PVPP power, and system imbalance, we identified three candidate time windows for providing counter-imbalance. Next, a series of experiments were conducted to explore how the different installation parameters influence the total energy cost of charging EVs. To derive a basic insight, we started with evaluating the baseline scenario without providing counter-imbalance, which we used as a reference case. This was compared with scenarios involving optimized energy flows together with counter-imbalance speculations at different levels of SI prediction accuracy.

The results indicate that, for the baseline scenario, the battery capacity and the installed PVPP power have the most significant impact on the total energy costs. A higher BESS capacity leads to higher costs. This is due to similar amounts of manipulated energy across the scenarios, as the PVPP production and EV consumption remained the same to maintain the comparability of experiments. Thus, additional capacity gets mostly unused, but it increases the costs. Regarding the PVPP, the balance between the generated cost and profits from sales to the grid depends on the time window. In the morning time window, the minimal PVPP installed power is sufficient as solar irradiation is low, and the profit is generated from obtaining energy together with the reward from the counter-imbalance in some periods and sending the energy back to the grid in some other periods when the imbalance settlement price favours deliveries to the grid. This effect is supported by the ascending trend of the DAM, as shown in Figure 5, and close-to-zero EV charging demand. In the remaining time windows, it is the opposite. Larger PVPP installed power leads to lower costs as the solar irradiation is higher. In the scenario with providing counter-imbalance, the accuracy of the SI prediction proved to be the key parameter. The level of prediction accuracy at which the system is able to achieve lower total energy costs compared to the baseline scenario lies between 80 and 90%. Generally, minimizing the risk and setting a reasonably low SI speculation limit is advantageous because the risk of a penalty with a combination of high SI speculation limits increases costs.

We found that SI speculation can reduce costs in installations equipped with BESSs by providing counter-imbalance during periods of minimal electric vehicle charging demand. However, it is essential to configure the individual installation component parameters properly. Charging infrastructure operators should not only consider investments in developing high-quality prediction models, but such models should also be used together with more sophisticated speculation strategies. Furthermore, using technologies with lower initial and operation costs can help elevate the benefit of SI speculation.

*Outlooks and Further Implications*

Further research will focus on the development of SI prediction models. Combining predictions with estimated prediction reliability levels opens the possibility for more complex SI speculation strategies. Speculations could be limited to periods with low risk to strategically select multiple time windows or more actively respond to the currently available energy instead of only following plans proposed by an optimization model. A deeper analysis of EV charging behaviour and the use of more sophisticated localized weather forecasts could also be an area for improvement. A comprehensive economic analysis could be another subject of further research.

In this study, we considered residential charging stations. For commercial charging stations, the energy consumption patterns are more random. The applicability of the proposed approach is strongly tied to the predictability of energy flows. Thus, applying the proposed approach to commercial charging stations needs to be studied separately.

The exploitation of counter-imbalance depends on several factors. An important factor is the related legislation and regulations, which set the rules for financial incentivization of counter-imbalance. Organizations responsible for their own imbalance must dedicate additional expenses to implement energy management systems that incorporate similar methodologies to the one proposed in this paper. Transmission and distribution system operators positively value local flexibility, including the provision of counter-imbalance. However, nowadays, counter-imbalance speculations are not a usual practice in Slovakia, and no specific regulations address them. With the expected increase in the share of flexible devices, higher motivation for counter-imbalance speculations can also be expected. Hence, policymakers may need to introduce a legislation framework regulating counter-imbalance speculations.

**Author Contributions:** Conceptualization, P.B. and Ľ.B.; Methodology, Ľ.B.; Software, M.T. and M.S.; Validation, M.T.; Formal analysis, M.S.; Investigation, M.T.; Resources, D.M.; Data curation, M.S.; Writing—original draft, M.S.; Writing—review & editing, P.B. and Ľ.B.; Visualization, M.T.; Supervision, P.B. and Ľ.B. All authors have read and agreed to the published version of the manuscript.

**Funding:** This work was supported by the project IPCEI_IE_FLOW_BESS_012021, which received funding from the call OPII-MH/DP/2021/9.5-34 under the grant 313012BLP2.

**Data Availability Statement:** Data is unavailable due to privacy reasons. Please contact authors to get access to the used data.

**Conflicts of Interest:** Authors Marián Tomašov, Milan Straka, Peter Braciník and Ľuboš Buzna were employed by the company INO-HUB Energy j.s.a. The remaining authors declare that the research was conducted in the absence of any commercial or financial relationships that could be construed as a potential conflict of interest.

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
