# Peer review of "A Feasibility Study of Profiting from System Imbalance Using Residential Electric Vehicle Charging Infrastructure"

_energies, doi:10.3390/en16237820_

Round 1

Reviewer 1 Report

Comments and Suggestions for Authors

The article is devoted to a feasibility study for profiting from system imbalance in the charging infrastructure for household electric vehicles. Computer simulations were used to estimate total operating costs by varying key plant parameters and the accuracy of the prediction model.

However, there are some comments regarding the work:

1. The Abstract section must be rewritten to reflect the relevance of the problem being solved and the scientific novelty of the solution obtained.

2. Keywords must be adjusted, highlighting special terms that characterize the study.

3. In the Introduction section, the relevance of the research being conducted should be indicated.

4. How was equation (7) obtained?

5. It should be explained in what interval the forecast for photovoltaic generation in paragraph 2.6 was made.

6. The list of cited sources should include more modern publications on algorithms for the operation of energy systems, including renewable generation sources, for example,

https://doi.org/10.3390/math11153317

https://doi.org/10.3390/app12062905

https://doi.org/10.3390/math11102394

7. Figure 3 shows an energy flow diagram to which it would be appropriate to reduce the energy balance equation of the network, including losses in it.

8. A Discussion section should be added, in which it is necessary to characterize the models obtained and the scientific results obtained, describe their advantages and disadvantages, and also provide the limitations of the proposed energy balance model.

9. Conclusions must be structured, highlighting the main scientific and practical results obtained, supporting them with numerical results.

Reviewer 2 Report

Comments and Suggestions for Authors

The study explores the potential of battery storage systems in residential chargers to address grid imbalances. The authors use an optimization model that identifies proper time windows and parameters where battery can effectively reduce costs and imbalance.

The manuscript is interesting and it is generally well-written. The presented results can provide useful insights into the energy management and ancillary service topic. However, there are some concerns that need to be adequately addressed:

1. The integration of multiple residential chargers with BESS into the grid can affect grid stability, especially during peak demand times and potential issues like reverse power flow. How about this consideration?

2. Why specify this method for residential EV infrastructures? If the proposed model were scaled up for industrial or commercial EV charging infrastructures, how might the dynamics change or what additional challenges may arise?

3. Energy management models including EVs, batteries, and renewable generation often have various uncertainties. These uncertainties extensively discussed in many recent studies such as https://doi.org/10.1109/SMC42975.2020.9283440 and https://doi.org/10.1109/TSG.2022.3172726   (references that could be discussed and cited in the paper). How the presence of uncertainty in the real systems such as variable weather conditions, prices, or consumer behavior like inconsistent EV charging times or energy consumption patterns can impact the efficiency of the method?

4. Table 4 mentions the number of BESS cycles. Were there any considerations about the degradation of BESS over time due to charge and discharge cycles?

5. Abbreviations should be defined with their full terms before their first use. For example,in the Abstract, BESS and EV are used without specifying their full forms (also more: SI rewards, SI speculations, etc.).

6. Using contractions such as " It's very interesting and encouraging for …" in formal writing should be avoided.

Comments on the Quality of English Language

The manuscript requires minor editing for English. There are some typo and grammatical mistakes that need to be addressed.

Reviewer 3 Report

Comments and Suggestions for Authors

Dear Authors, this paper investigates profiting from system imbalances on residential electric vehicle charging infrastructure. I want to express my appreciation for the effort you have put into your research. However, there are some comments and suggestions to improve the quality of the manuscript. The comments are as follows:

·         The abstract might benefit from a bit more clarity regarding the main findings and their implications.

·         In the introduction it is important to make a clearer statement of the research problem or the research questions.

·         The introduction could potentially provide more context regarding the current state of residential chargers and their impact on grid stability, to establish a strong foundation for the reader. Furthermore, the objectives of the study have to be clearly stated and be directly tied into the problem statement.

·         Consider providing a clearer synthesis of existing literature, highlighting gaps that the present study aims to address. Moreover, ensure that the transition from the literature review to the main study is smooth and logical. Suggested literature:

o   https://doi.org/10.1016/j.segan.2022.100620

o   https://doi.org/10.1109/SmartGridComm.2019.8909689

o   https://doi.org/10.1109/TSG.2014.2336623

o   https://doi.org/10.3390/en13225858

o   https://doi.org/10.3390/en14040866

·         Ensure that the concept of counter-imbalance is explained with clarity and simplicity for a wide range of readers.

·         Providing a comparative analysis of the different scenarios and experiments conducted, highlighting the strengths and weaknesses of each would strongly enhance the quality of the paper.

·         In the conclusions, provide clear and actionable recommendations for future research, policy, or practice based on the findings. Finally I’d suggest to discuss the broader implications of the findings, considering various stakeholders such as policymakers, practitioners, and the academic community.

Round 2

Reviewer 1 Report

Comments and Suggestions for Authors

In general, the authors revised the article. But before publication, authors should prepare a literature document according to the journal’s rules.

Reviewer 2 Report

Comments and Suggestions for Authors

The authors have responded to some of my comments sufficiently, but some critical points still remain unclear to me:

1. There seems to be a misunderstanding about Comment 1. Although the use of BESS can help grid stability from one perspective,it is important to look at the possible downsides especially when adding many chargers and inverters to the grid alongside BESS. This point is important because the focus of the paper is on grid stability. The charging and discharging cycles of BESS combined with variable residential loads can lead to voltage fluctuations in the grid. The integration of BESS and inverters can also introduce harmonics into the power system and affect powr quality. If multiple residential chargers are charging simultaneously, this can lead to an overload of the local distribution networks and careful management of grid capacity is necessary.

2. Description to Comment 3 is also not sufficient. The role of uncertainty is crucial as it is directly linked to stability issues. The paper should discuss how the method addresses these aspects. Furthermore, it is important to include and properly discuss more references including the suggested ones.

3. Also, many revisions with track changes have made the revised manuscript cluttered and difficult to track the main revised concepts.

Comments on the Quality of English Language

Some typos and grammatical mistakes should be addressed.

Reviewer 3 Report

Comments and Suggestions for Authors

Dear Authors, thanks for addressing my comments. No further comments from my side.

Round 3

Reviewer 2 Report

Comments and Suggestions for Authors

The authors have adequately addressed my comments, and the manuscript has improved significantly. I believe it is now suitable for publication in the journal.